# Clinical Assessment, Genetics, and Treatment Approaches in Autism Spectrum Disorder (ASD)

**DOI:** 10.3390/ijms21134726

**Published:** 2020-07-02

**Authors:** Ann Genovese, Merlin G. Butler

**Affiliations:** Department of Psychiatry & Behavioral Sciences, University of Kansas Medical Center, Kansas City, KS 66160, USA; agenovese@kumc.edu

**Keywords:** autism, ASD, genetics, heterogeneity, syndromes, assessment, medications, treatment, causes

## Abstract

Autism spectrum disorder (ASD) consists of a genetically heterogenous group of neurobehavioral disorders characterized by impairment in three behavioral domains including communication, social interaction, and stereotypic repetitive behaviors. ASD affects more than 1% of children in Western societies, with diagnoses on the rise due to improved recognition, screening, clinical assessment, and diagnostic testing. We reviewed the role of genetic and metabolic factors which contribute to the causation of ASD with the use of new genetic technology. Up to 40 percent of individuals with ASD are now diagnosed with genetic syndromes or have chromosomal abnormalities including small DNA deletions or duplications, single gene conditions, or gene variants and metabolic disturbances with mitochondrial dysfunction. Although the heritability estimate for ASD is between 70 and 90%, there is a lower molecular diagnostic yield than anticipated. A likely explanation may relate to multifactorial causation with etiological heterogeneity and hundreds of genes involved with a complex interplay between inheritance and environmental factors influenced by epigenetics and capabilities to identify causative genes and their variants for ASD. Behavioral and psychiatric correlates, diagnosis and genetic evaluation with testing are discussed along with psychiatric treatment approaches and pharmacogenetics for selection of medication to treat challenging behaviors or comorbidities commonly seen in ASD. We emphasize prioritizing treatment based on targeted symptoms for individuals with ASD, as treatment will vary from patient to patient based on diagnosis, comorbidities, causation, and symptom severity.

## 1. Introduction

Leo Kanner in 1943 [1] first introduced the term autism as a diagnostic label to define a specific syndrome observed in young children manifested by early onset, characteristic symptomatology, and disrupted social and emotional relationships. Since then, autism is now recognized as Autism Spectrum Disorder (ASD), which is classified as a developmental disorder as defined in DSM-5 (Diagnostic and Statistical Manual of Mental Disorders, 5^th^ Edition) by the American Psychiatric Association [2] and the ICD-10 (International Classification of Diseases, 10^th^ Revision) by the World Health Organization [3]. Autism is characterized by significant impairment in social communication and atypical repetitive and/or restrictive behaviors or interests, with an onset in the early developmental period, prior to age 3 years. The American Academy of Pediatrics [4] recommends screening all infants and toddlers to identify early signs of autism at 18 months and again at 24 months of age. Rating or assessment scales that have been validated for both clinical and research purposes are helpful in establishing the diagnosis of autism. These scales include the Autism Diagnostic Interview-Revised (ADI-R) and the Autism Diagnostic Observation Schedule, Second Edition (ADOS-2) and should be administered by trained specialists in conjunction with an evaluation of the child with consideration of history and clinical presentation [5,6].

ASD affects between 1 to 2% of children in United States with a growing role for genetic factors with etiological heterogeneity. ASD can be conceptualized as a behavioral syndrome rather than a specific categorical mental disorder [7]. The concept of “syndromic autism” (ASD associated with morphological signs or symptoms helpful in the identification of specific genetic disorders) stands in contrast to “non-syndromic autism” (idiopathic ASD with no associated signs or symptoms). Multiplex autism refers to those with a positive family history of other similarly affected individuals, which highlights the heterogeneity of ASD [8,9].

Clinical and other health concerns that may be associated with ASD include intellectual disability (ID), electroencephalogram (EEG) abnormalities with or without epilepsy, dysmorphic features, and abnormal MRI findings [10,11]. About 10% of children with autism are reported to have microcephaly [12,13], which may be associated with additional findings and a poor prognosis. On the other hand, a large-appearing head size is common in children with autism along with increased brain volumes, particularly in the frontal lobes, but with smaller occipital lobes [14,15,16,17,18,19]. Mutations of the phosphatase and tensin homolog (PTEN) tumor suppressor gene were reported by Butler et al. [14] in children with autism and extreme macrocephaly. Recent studies have shown that about 20% of genes implicated in autism are also known cancer genes, thereby stimulating an interest in not only risks for cancer development in individuals with ASD but whether chemotherapeutic agents could play a role in treatment of autism [20].

Tordjman et al. [7] provided a comprehensive review of diverse genetic disorders associated with autism and considered possible common underlying mechanisms leading to a similar cognitive-behavioral phenotype of autism, while examining relevant genetics, syndromes, epigenetics, and environmental factors. Despite the recognition of nearly 800 susceptibility, clinically relevant, or known genes for autism spectrum disorder collated by Butler et al. [21] and characterized by numerous etiological studies including relevant animal models [22], it appears that no cohesive model of causation, biomarker [23], or specific mode of transmission for the development of autism has been firmly identified [24].

The cause of ASD is heterogenous involving genetics with multiple different gene variants and environmental influences triggering physiological changes in genetically sensitive individuals along with in utero and metabolic factors including mitochondria dysfunction reported in 10 to 20% of patients with ASD. Familial and heritability studies have shown that genetic factors contribute, with estimates as high as 90% with tuberous sclerosis, fragile X, and Rett syndromes as examples of single gene conditions found but accounting for less than 10% of all ASD cases [25,26,27]. A list of genetic syndromes and chromosomal disorders associated with ASD is illustrated as Box 1 below.

Behavioral and psychiatric comorbidities are common in individuals on the autism spectrum, and can have a substantial impact on overall health, quality of life, and long-term prognosis. Approximately 30% of individuals with ASD require psychological and psychiatric treatments including medication for behavioral problems including hyperactivity, impulsivity, inattention, aggression, property destruction, self-injury, mood disorders, and psychotic or tic disorders [28,29], a major focus of our report. 

Box 1A List of Genetic Syndromes and Chromosome Findings where Autism is a Recognized Feature.
Adenylate succinase deficiency

Myotonic dystrophy

Angelman and Prader-Willi syndromes (AS–maternal or PWS–paternal 15q11-q13 deletions)

Neurofibromatosis

Apert syndrome

Noonan syndrome

15q11.2 BP1-BP2 microdeletion (Burnside-Butler) syndrome

Oculo-auriculo-vertebral spectrum

CHARGE syndrome

Phelan-McDermid syndrome (22q13 deletion)

Chromosome 15 duplications (maternal origin)

PTEN gene associated disorders with extreme macrocephaly (Cowden/Bannayan-Riley-Ruvalcaba syndrome)

Chromosome 16p11.2 deletions

Rett syndrome (MECP2 gene)

Cohen syndrome

Shprinzten/velo-cardio-facial/DiGeorge(22q11 deletion)

De Lange syndrome

Smith-Lemli-Opitz syndrome

Down syndrome

Smith-Magenis syndrome (17p11.2 deletion)

Duchenne muscular dystrophy

Sotos syndrome

Fragile X syndrome (FMR1 gene)

Tuberous sclerosis

Hypomelanosis of Ito

Turner syndrome

Joubert syndrome

Untreated or poorly treated phenylketonuria (PKU)

Mitochondrial dysfunction

Williams syndrome

Moebius sequence

Modified from G.B. Schaefer and N.J. Mendelsohn, “Genetics evaluation for the etiologic diagnosis of autism spectrum disorders,” Genetics in Medicine, vol. 10, pp 4–12, 2008 [25] and from M.G. Butler and others, “Assessment and treatment in autism spectrum disorders: A focus on genetics and psychiatry”, Autism Research and Treatment, vol. 2012, 242537, 2012 [30].

## 2. Diagnosis and Genetics of ASD

ASD affects about 1 individual in 50–100 live births [31,32] and is on the increase with a higher prevalence than reported for congenital brain malformations or Down syndrome. The recurrence rate may be as high as 25–30% if a second child is also diagnosed with ASD in a family (i.e., multiplex) compared with a sporadic pattern (simplex) form of ASD. High heritability estimates have been reported in ASD, e.g., 70 to 90% concordance rate in monozygotic twins [33,34], indicating the potential importance of genetics, but studies have not identified the anticipated number of pathogenic variants to date. Those without a family history may be at a greater risk of copy number variants (CNVs) or deletions/duplications at the chromosome level using chromosomal microarray analysis and DNA probes for CNVs and comparative genomic hybridization [35]. Furthermore, 10% of the individuals with autism from simplex families had CNVs, while only 3% of individuals with autism from multiplex families with more than one family member affected showed CNVs, compared with 1% seen in normally developing children studied as controls. The majority of CNVs were of the deletion type. Single gene conditions are found in about 20% of subjects with ASD, while epigenetics impacted by environmental factors such as nutrition, infections, or toxins could alter the gene status through methylation, controlling function without changing the DNA sequence [27,36,37]. Genome-wide linkage and association studies (GWAS) have identified hundreds of ASD risk gene loci in all human chromosomes.

Autism is considered the most heritable neurodevelopmental disorder based on a large difference in concordance rates or heritability estimates between monozygotic and dizygotic twins with monozygotic twins having rates that are nearly three times higher than rates found in dizygotic twins [38]. Furthermore, a meta-analysis of twin studies on the heritability of ASD in more than 6000 twin pairs was reported by Tick et al. [39]. They found that correlations for monozygotic twins were very close to perfect at a score of 0.98 while the score for dizygotic twins was 0.53, indicating a role of shared environmental effects. Hallmayer et al. [38] concluded that susceptibility to ASD showed moderate genetic heritability and substantially shared twin environmental components, indicating a challenge to find genetic causation for autism.

Genetic investigations have identified the role of hundreds of gene variants, but risk effects are highly variable and relate to other conditions besides autism, making it difficult to find ASD-specific gene variants [40,41,42]. Many gene variants do impact on common biological pathways or interactions and may play a potential causative role in autism, but more research is needed to address the current challenges in translating autism genetics into clinical practice as genetic etiology and pathogenesis of ASD remain largely unclear [43,44,45].

Further advances made in genetic technology and testing with improved DNA sequencing and development of bioinformatics with searchable computer genetic variant databases have led to discoveries and characterization of genetic defects in the potential causation of ASD. Improvements of chromosome microarray technology with combination of probes for both copy number variants and single nucleotide polymorphisms (SNPs) have not only led to enhanced testing capabilities in identifying segmental deletions and duplications in the genome, but also the identification of pathogenic or disease-causing genes and their positions within chromosomal regions.

### 2.1. Genetic Factors Contributing to Autism

Advances in genetic testing and evaluation for syndromic causation of patients with ASD have identified an etiology in up to 40%, using a three-tier clinical genetic approach described by Schaefer and others in 2008 [25] and later in 2013 [34] to identify causes in children diagnosed with ASD. These include fragile X, Rett, and other genetic syndromes, such as tuberous sclerosis (10–20%), PTEN gene mutations (3%), and structural chromosomal deletions or duplications using early versions of chromosomal microarrays (3%), and an additional 10% or higher when using high-resolution microarray technology. Metabolic disorders such as mitochondrial dysfunctions are seen in 10 to 20% of patients with ASD [32,34,46]. Children with ASD reported with microdeletions or duplications involve chromosome regions 1q24.2, 2q37.3, 3p26.2, 4q34.2, 6q24.3, 7q35, 13q13.2-q22, 15q11-q13, 15q22, 16p11.2, 17p11.2, 22q11, 2q13, and Xp22 [13] and additional cytogenetic disorders associated with ASD are found with new ultra-high-resolution microarray technology (e.g., 15q11.2 BP1-BP2 deletions) [47]. Recent GWAS findings in ASD and broad autism phenotype in 28 extended pedigrees from Canada and the United States showed additional chromosome regions including 1p36.22, 2p13.1, 6q27, 8q24.22, 9p21.3, 9q31.2, 12p13.31, 16p13.2, and 18q21.1 [48].

These newer chromosomal SNP microarrays can identify abnormalities 100 times smaller than can be seen with high-resolution chromosome methods including for ASD candidate genes. A report by Shen et al. [26] on 933 patients with ASD using standard karyotype analysis, fragile X DNA testing, and chromosomal microarrays found abnormal karyotypes in 2.2%, abnormal fragile X testing in 0.5%, and microdeletions or microduplications in 18.2% of subjects. These included recurrent deletions or duplications of chromosome 16p11.2 [49] and for chromosome 15q13.2q-13.3, while new studies found chromosome 7q11, chromosome 15q11.2 BP1-BP2, and chromosome 22q11.2 [50].

Whole-exome sequencing (WES) have identified yields of up to 30% [51] but other studies show lower results (e.g., 9.3%) in individuals with ASD [52]. The vast majority of gene variants are of uncertain clinical significance due in part to the rarity found in genomic normative datasets and limitations of bioinformatics, evolutionary conservation, computational predictions, and relevance in relationship to the normal population. Likely explanations for the lack of consistency among molecular diagnostic testing results may relate to multifactorial causation of ASD influenced by a complex interplay between inheritance and environmental effects along with contributions by epigenetics on gene expression. Despite considerable interest in identifying autism-specific genes, deleterious variants have been implicated across multiple neurodevelopmental and psychiatric disorders but insufficient to date in identifying those genes that, when mutated, confer a largely ASD-specific risk [42].

An early genome-wide association study (GWAS) on 4300 affected children with ASD reported by Wang et al. [53] and 6500 controls of European ancestry found a strong association with six single nucleotide polymorphisms (SNPs) located between cadherin 10 (CDH10) and cadherin 9 (CDH9) genes located on chromosome 5 encoding neuronal cell-adhesion molecules. Since then, over 100 genetic loci have been reported to be associated with ASD [54,55], comprising genes converging on chromatin–remodeling, synaptic function in neuronal signaling, and neurodevelopment [56,57]. Furthermore, Butler et al. [21] collated about 800 genes from the literature that were implicated as clinically relevant, susceptible, or known in ASD. These multiple genes include several members of the neuroligin, neurexin, GABA receptor, cadherin, and SHANK gene families. Other genes were found to code for neurotransmitters and their receptors, transporters, oncogenes, brain-derived hormones, epigenetics, and signaling and ubiquitin pathway proteins, along with neuronal cell-adhesion molecules [21,58,59,60,61]. 

### 2.2. Metabolic Factors Contributing to Autism

Metabolic factors are now recognized as contributing to autism including the mitochondria. Next-generation DNA sequencing now allows for accurate detection of mutations or gene variants at the nuclear and mitochondrial DNA (mtDNA) level and is potentially more informative than chromosomal microarray analysis involved in structural DNA changes. This technology is now available in the clinical setting for individuals presenting with biochemical and mitochondrial disturbances and autism [21,62]. Three functional pathways to ASD are potentially involved, which include genes and pathways for chromatin remodeling, (e.g., CHD7, MECP2, DNMT3A, and PHF2), Wnt (e.g., CHD8, PAX5, and ATRX), and other signaling super-pathways (e.g., GPCR, ERK, RET, and AKT) [50,51] and mitochondrial dysfunction in ASD (e.g., [62]). High lactate levels are also reported in about one in five children with ASD, further supporting the role of the mitochondria in energy metabolism and brain development [32,46,62].

The mitochondria are intracellular organelles found in the cytoplasm which play a crucial role in adenosine 5’-triphosphate (ATP) production through oxidative phosphorylation [62,63,64,65,66], the latter process carried out by the electron transport chain made up of Complexes I, II, III, and IV situated in the inner membrane of the mitochondria containing about 100 proteins. Genes that encode the proteins are located in both nuclear and mitochondrial DNA [65,66,67] and are required for cellular energy that can impact or influence brain development and activity. There are hundreds of nuclear genes involved in mitochondrial function, while only 13 mitochondrial genes code for protein. Mitochondrial disturbances include a depletion type, or reduced number of mitochondria per cell, with a decreased quantity of mtDNA, or mtDNA mutations producing defects in biochemical reactions within the mitochondria and individual cells [32,46,65,68].

A subset of individuals with ASD can have small mitochondrial DNA deletions/duplications detectable with mitochondrial genome microarrays. Human mitochondrial DNA (mtDNA) is a circular, double-stranded DNA molecule contained within the mitochondrion and inherited solely from the mother. Each mitochondrion contains 2–10 mtDNA copies. In humans, 100–10,000 separate copies of mtDNA are usually present per cell [63,64,65]. Inborn errors of metabolism may contribute significantly to the causation of ASD with enzyme deficiencies leading to an accumulation of substances that can cause toxic effects on the developing brain. A common example is phenylketonuria, leading to excessive phenylalanine levels, intellectual disability, and ASD, if not diet controlled. 

## 3. Clinical Assessment and Testing

### 3.1. Initial Clinical Evaluation

A healthcare professional interviews the parent or caregiver regarding presenting problems, reviews a three-generation family history, developmental milestones, and abnormal behaviors of the child, medical and surgical history, and any past or current treatments. The diagnostic evaluation is typically performed by a developmental pediatrician or a child and adolescent psychiatrist. Physical and mental status examinations are performed, and additional testing ordered, as appropriate. If a positive family history for autism is found or dysmorphic (syndromic) features, then a referral is made for clinical genetics’ evaluation. Laboratory evaluations may include genetic testing, lead levels, thyroid function, lactate, pyruvate and cholesterol levels, and urine for organic acids. Referrals are made for neurological evaluations and brain imaging when clinically indicated.

### 3.2. High-Resolution Microarrays and ASD

Genetic testing often begins with chromosomal microarray analysis (CMA) to identify copy number variants (CNVs) to search for a cause of autism spectrum disorder and other related conditions. Microarrays employ a variety of designs and range of coverage of genomic regions, which increases the diagnostic yield as arrays have evolved over time to include better coverage and accuracy. Often the CNVs identified are unclassified or poorly understood in their role in causation of ASD.

Neurodevelopmental disorders can cumulatively affect up to 15% of children [69]. While the etiology of ASD is complex, it involves genetic factors with 800 genes recognized, accounting for 4% of all human genes that are implicated in ASD [21]. Single gene changes, large genomic structural changes (i.e., deletions or duplications), or smaller CNVs and other polygenic conditions can be influenced by the environment and epigenetic factors [70,71]. Genetic testing to pinpoint the underlying cause of ASD is critical for clinical management and counseling. Further, chromosomal microarray analysis has demonstrated the highest diagnostic yield for individuals with ASD as compared to other genetic tests as well as in individuals with ID and/ or behavioral problems, including in developing countries.

High-resolution microarrays now utilize millions of single nucleotide polymorphisms (SNPs) as probes to test the DNA from patients presenting with neurodevelopmental disorders, intellectual disabilities, and ASD. These SNP microarrays are used to identify microdeletions (or duplications) with recognition of dozens of a growing list of deletion or duplication syndromes not previously detected. For example, a study of custom-made, ultra-high-resolution microarrays reported by Ho et al. [47] in 2016 were optimized for the detection of neurodevelopmental disorders (Lineagen, Salt Lake City, Utah) on 10,351 patients presenting for genetic services for neurodevelopmental disorders, ASD, ID, behavioral problems, or with or without multiple congenital anomalies (MCA) over a period of four years. Their testing sample had a male:female ratio of 2.5:1 with a mean age of 7 years. Fifty-five percent of cases represented patients with a diagnosis of ASD with or without other features. The overall CNV detection rate of 28.1% was seen in 10,351 consecutive patients and 24.4% in those with ASD along with 33% in those with intellectual disabilities and/or MCA without autism. The rate of pathogenic findings was significantly lower (4.4%) when the diagnostic indication was ASD only compared to diagnostic indications of DD/ID/MCA without a reported diagnosis of ASD (i.e., non-ASD cohort) (12.5%).

In the ASD cohort, the overall pathogenic rate was slightly higher for individuals with ASD+ as compared to the overall pathogenic rate for individuals with ASD only. The pathogenic rate in the ASD+ cohort started at 4.1% in the youngest group and rose to 8.5% in the 5.5–10 years range. The pathogenic rate in the ASD only cohort rose gradually with age, from 3.4% in the youngest cohort (0–3.4 years) to a peak at 7.0% in adolescence. In 5694 patients classified as ASD and 4657 patients with non-ASD, the most common findings were 15q11. 2 BP1-BP2 deletions followed by proximal 16p11.2 deletions or duplications, 15q13.3 deletions, and 16p13.1 duplications (see Figure 1). The most common finding in the non-ASD cohort was the 22q11.2 deletion causing velo-cardio-facial or DiGeorge syndrome. This study illustrates the value of CMA testing and its impact on medical management is now recognized in consensus medical guidelines for the evaluation of children with ASD.

### 3.3. Next-Generation Sequencing (NGS)

Advances in genomics technology using next-generation sequencing (NGS) have led to discovery of many disease-causing genes using candidate gene approaches, disease-specific gene panels, or by whole-exome sequencing of patients presenting with neurodevelopmental disorders, intellectual disabilities, or ASD. Applying genomics to the study of neurodevelopment and function has identified over 5000 implicated genes using clinical exome sequencing approaches and informatics in affected individuals. In addition, disease-specific NGS gene testing panels have been developed and used in the commercial laboratory setting for testing patients presenting for genetic services, including approximately 600 genes for intellectual disabilities and over 100 genes available for testing for ASD (e.g., Fulgent Diagnostics, Irvine, California).

These types of analyses have identified pathogenic gene variants or mutations which are known to be disease-causing such as missense or nonsense, but more often variants of unknown clinical significance are found. More testing and information with better interpretations of the genomic change and impact at the protein level are needed to help determine the role, if any, of the unknown gene variants in causing the disease under study including for ASD. Hundreds of new causative genes relating to human diseases and syndromes have been identified with the use of NGS technology over the past few years, with expectations of continued success given improvements in genetic technology, bioinformatics, and expanded genomic databases to search for gene variants and in further characterizing identified genes.

Next generation DNA sequencing of the exons (referred to as exome sequencing) or whole-genome sequencing will continue to find new discoveries of disease-causing SNPs, gene regulatory sequences, or mutations of protein-coding genes for both structural and regulatory proteins. Identifying molecular signatures of novel or disturbed gene or exon expression, disease-specific profiles and patterns (i.e., expression heat maps), and recognition of interconnected gene pathways in autism and other behavioral disorders in the future by using readily available blood elements (e.g., lymphoblasts) should hold promise for treatments with pharmacological agents by regulating (either increasing or decreasing) activity of normal (or abnormal) gene function. The study of non-coding RNAs, which control the amount or quantity of gene expression coding for protein production through micro-RNAs and the quality of protein production by specific isoform development by sno-RNAs, will lead to new areas of research and medical therapies for human diseases. Therefore, this technology should be considered in the diagnostic evaluation of ASD, either sporadic or with a positive family history of others similarly affected.

Butler et al. [21] searched the literature and found approximately 800 genes implicated in autism in the literature as clinically significant, relevant, or known to contribute to the risk of ASD. Recent research revealed that ASD and cancer genes may share common genetic architecture and pathways with the first evidence of the PTEN tumor-suppressor gene playing a role in autism in 2005 [14]. Hence, approximately 800 ASD-related genes and 3500 genes in cancer were examined using the GeneAnalytics pathways and profiling software programs and found shared cell-signaling pathways, metabolic disturbances, and molecular functions in 138, or 17%, of ASD genes that overlap with cancer genes [20]. Shared mechanisms may lead to identification of common pathology and a better molecular understanding of causation as well as potential treatment options.

## 4. Treatment Approaches

### 4.1. Behavioral Interventions in ASD

#### 4.1.1. For Children and Adolescents with ASD

Weitlauf et al. [72] reviewed 65 studies, comprising 48 randomized trials and 17 nonrandomized comparative studies, that analyzed the benefit of behavioral interventions. High-intensity applied behavior analysis (ABA) was associated with improvement in cognitive functioning and language skills relative to community controls in young children [73]. Early intensive behavioral intervention (EIBI) is a well-established treatment for young children with ASD and is based on the principles of applied behavior analysis. Delivered over a period of several years at an average of 20 to 40 hours per week, it can provide substantial benefit for core ASD symptoms, particularly in terms of communication skills [74]. Social skills’ interventions including group administered training showed positive effects on social behaviors for older children [75].

#### 4.1.2. For Adults with ASD

National Institute for Health and Care Excellence (NICE) recommended guidelines for management and support of children and young people with autism using group or individual social learning programs to improve social interaction deficits by applying behavioral therapy techniques within a social learning framework. These include using video modeling, peer feedback, imitation, and reinforcement to teach conventions of appropriate social interpersonal interaction [76]. There is evidence from observational studies in adults with ASD that social skills’ groups may be effective at improving social interaction [77]. CBT can help adults with ASD across a range of domains, particularly in the context of treating anxiety and OCD, and supporting adults who have difficulties related to a history of victimization [78].

### 4.2. Medication Treatments in ASD

Psychopharmacological treatment of ASD is challenging due to considerable variability in the presentation of ASD and commonly occurring comorbidities. Individuals with ASD are typically more vulnerable to side effects of psychopharmacological agents than their age-matched, neuro-typically developing peers [79]. Finally, ASD impacts individuals over the course of their lifespan and most of the literature on psychotropic medications in ASD involves pediatric populations.

A psychopharmacological approach may be beneficial in the treatment of identified target symptoms in individuals with ASD. When considering the use of medications, potential benefits and risks must be weighed on a case-by-case basis. It has been reported that close to half of insured children with ASD are receiving psychopharmacological interventions, most commonly with stimulants, alpha-2 agonists, antipsychotics, anticonvulsants, and antidepressants [80].

Currently, there are no medications approved for treatment of the core symptoms of ASD including social communication deficits or repetitive behaviors. Common target symptoms for which there are effective, evidence-based medication treatment include hyperactivity, inattention, impulsivity, irritability, aggression, self-injurious behavior, repetitive behaviors (including stereotypies), and insomnia [81]. For the treatment of irritability associated with ASD, the antipsychotics risperidone and aripiprazole are licensed and approved by the US Food and Drug Administration [82].

#### 4.2.1. For the Treatment of ADHD Symptoms in ASD

Stimulant medications are considered first line agents for attention deficit hyperactivity disorder (ADHD) in individuals with ASD, given that overall they are most often effective and generally well tolerated compared to other ADHD medications. The RUPP research team [83] and later Reichow et al. [84] demonstrated a clear superiority of methylphenidate over placebo in children with pervasive developmental disorder. However, those with ASD had a greater risk of side effects with methylphenidate including decreased appetite, insomnia, depressive symptoms, irritability, higher levels of social withdrawal, and lower treatment response rates compared to youth with ADHD alone. It should be noted that stimulant medications for ADHD in the amphetamine class are often used in children with ASD but have not been as rigorously studied.

Regarding non-stimulant medications for ADHD in ASD, both atomoxetine and alpha-2 agonists have shown benefit. Harfterkamp et al. [85], in a double-blind treatment trial of patients age 6 to 17 years with ADHD and ASD using atomoxetine 1.2 mg/kg/day or placebo for eight weeks, found that atomoxetine moderately improved ADHD symptoms, but with frequent adverse events including nausea, decreased appetite, fatigue, and early morning awakening. The alpha-2 agonist guanfacine has been shown to be effective for ADHD in children with ASD demonstrated by a double-blind, placebo-controlled trial of guanfacine extended release in which 50% of youth on active treatment improved on the Clinical Global Impression–Improvement (CGI-I) scale [86], compared to 9.4% on placebo [87], with sedation and transient lowering of blood pressure as the most common adverse effects.

#### 4.2.2. For the Treatment of Irritability, Aggression, and Self-Injurious Behavior in ASD

Atypical antipsychotics compared to other medications have to date demonstrated the best evidence for the treatment of irritability in ASD. Risperidone in youth age 5 to 16 years with ASD [88] in three randomized, placebo-controlled trials showed an over 50% reduction in the irritability score of the Aberrant Behavior Checklist (ABC-I) irritability scale [89] and the magnitude of the response was greater when irritability was rated as moderate to severe [90].

Aripiprazole, the second antipsychotic approved by the FDA for the treatment of irritability associated with autism (in children between the age of 6 and 17 years), demonstrated significantly lower severity scores on the ABC-I and the CGI-I scales for subjects on active medication compared to placebo in two large-scale, randomized, placebo-controlled studies. Unfortunately, weight gain is a common side effect of antipsychotics, and increases in body mass index have been shown to be similar for aripiprazole and risperidone in children with ASD [91].

Anticonvulsant medications divalproex and topiramate have shown some promise for treating irritability in ASD. Divalproex was beneficial in reducing irritability in a small, randomized, placebo-controlled trial of children with ASD [92]. However, an earlier trial failed to show separation from placebo on the ABC-I [93]. Topiramate as monotherapy has no demonstrated benefit in the treatment of irritability in youth with ASD [94]; however, it reduced the ABC-I score when co-administered at an average daily dose of 200 mg with risperidone [95]. It is hypothesized that, as EEG abnormalities are common in children with ASD, symptom reduction with anticonvulsants may result from treatment of abnormal brain discharges [96].

#### 4.2.3. For the Treatment of Repetitive Behaviors Including Stereotypies in ASD

Fluoxetine has been shown to improve repetitive behaviours in adults with ASD [97]; however, benefit has not been reliably demonstrated in pediatric populations. In fact, the Cochrane Collaboration published a systematic review [98], which concluded that for repetitive behaviors in children with ASD there is not only a lack of available evidence of benefit from treatment with selective serotonin reuptake inhibitors (SSRIs) including fluoxetine, fluvoxamine, and citalopram, but some evidence for risk of harm, given a greater incidence of adverse effects, most notably symptoms of behavioral activation.

#### 4.2.4. For the Treatment of Persistent Insomnia in ASD

Exogenous melatonin, (available as an over-the-counter supplement) in both immediate-release and extended-release formulations, has been shown to be safe and effective in improving sleep patterns in children with ASD [99]. Some evidence suggests that children with ASD have abnormal melatonin secretion and circadian rhythm abnormalities compared to non-ASD children [100]. Clonidine (an alpha-2 agonist) has shown promise in reducing latency of sleep initiation and decreasing nighttime awakening in ASD [101].

### 4.3. Pharmacogenetics and Role in Medication Selection and Management

Personalized or precision medicine is emerging in clinical practice based on individual genetic patterns contributing to pharmacogenetics, particularly in the field of psychiatry and treating individuals with ASD [102,103,104,105] with behavior issues including ADHD, irritability, aggression and self-injury, repetitive behaviors, and persistent insomnia addressed above. Pharmacogenetics is a study of structural DNA variation that impacts drug metabolism [106,107] and most often based on the cytochrome P450 enzyme system, primarily active in the liver and coded by genes. Cytochrome P450 enzymes metabolize or break down drugs in the liver with most prescription drugs metabolized by this enzyme system and, thus, play a significant role in the treatment of diseases [107,108]. Variation in drug response among individuals due to metabolism differences is now recognized as a major clinical problem, especially given that the use of several medications per patient is common practice. Relevant cytochrome P450 gene polymorphisms and different racial distributions can identify sources of variability in drug response by the modulation of metabolism by the cytochrome P450 enzymes impacting treatment in ASD.

There are over 50 cytochrome P450 hepatic enzymes that are primarily found in the mitochondria [102,103,104,105]. These enzymes metabolize endogenous and xenobiotic substrates including environmental pollutants and agricultural and plant-based chemicals and are involved in biosynthesis and metabolism of steroids, vitamins, hormones, lipids, and prostaglandins. About 90% of all drugs are metabolized by seven different cytochrome enzymes including CYP1A2, CYP3A4, CYP3A5, CYPC19, CYP2D6, CYP2C9 and CYP2B6 [106,109,110,111]. The most commonly prescribed medications used in treating patients with psychiatric problems and ASD are broken down by CYP2D6 [102,103,104,105]. It should also be noted that many drugs are also metabolized by more than one cytochrome P450 enzyme and in addition some drugs (e.g., risperidone) require break down to generate an active metabolite or functional agent for treatment.

There is growing evidence that cytochrome P450 enzymes may be altered by the environment in the form of inhibitors or inducers as well as impacting drug–drug interactions. Known inhibitors or inducers may include common sources such as caffeine, grapefruit, broccoli, cabbage, or cauliflower by impacting the individual enzyme activity. For example, if an individual has a reduced form of a cytochrome P450 enzyme, then an inducer may increase the enzyme response in breaking down the drug to help that person in metabolizing a specific medication and, thus, impact response to treatment.

Drug–drug interactions and their concentrations and half-life along with response to inhibitors and/or inducers can all impact medication levels and treatment in the patient. It should also be noted that individuals who are either fast or slow metabolizers based on their microsomal P450 enzyme system genotype patterns may respond differently to specific medications and put them at risk for either failure of drug therapy and/or adverse side effects. Similarly, a better understanding of the metabolic differences that occur with age will further impact on drug dosage and selection of specific therapeutic agents. Therefore, personalized medicine requires the development of resources for clinicians including pharmacogenetic dosing guidelines for medications, as 25 to 50% of individuals do not respond normally to drug dosage or treatment, and this scenario also applies to those with ASD [102].

Applying this knowledge from pharmacogenomics and identifying genes and polymorphisms involved in drug metabolism will benefit patients treated for psychiatric and behavioral problems. The discovery of new classes of drugs and research on existing drugs for new purposes to treat behavioral problems in patients with ASD are under investigation including clinical trials (e.g., in fragile X syndrome), holding promise for improved therapy. In addition, the discoveries made in brain imaging such as functional MRI or PET scans in identifying regions of the brain that are affected in ASD should allow for new treatment discoveries and applications specific for the altered regions identified.

## 5. Future Directions

Advances and application of genomic testing technology, bioinformatic approaches, and computational predictions will strengthen genetic testing results and interpretations as more experience is gained in testing patients presenting for clinical services and diagnosis [112,113,114]. Increased number of next-generation sequencing (NGS) or whole-exome sequencing (WES) studies in ASD of individuals from different ethnic backgrounds will be required to gain ASD-specific genomic information from datasets of both sexes when compared to the normal population. These contributions should allow a better understanding of the role of genetics, genomics, epigenetics, and specific candidate genes and their variants, along with environmental factors playing a role in ASD in relationship to multifactorial influences in family studies [38,112,114,115]. Confounding effects of clinical heterogeneity and diagnostic uncertainty are other complicating issues needing further characterization and evaluation to gain more experience in clinical assessment, genetics, and treatment approaches in autism spectrum disorder.

In addition, research with brain and tissue harvested and stored for structural DNA and RNA expression studies are needed on individuals with ASD having data from cognitive, behavioral, and ASD assessment tools and neuroimaging results while living. Coding and non-coding expression patterns and epigenetic (methylation) signals supplemented with WES and CNV data would be beneficial for a better understanding of the role of genetics in ASD, particularly with larger cohorts of individuals having similar genetic backgrounds, patterns, and ethnicity to identify large-effect pathogenic variants for facilitating genotype–phenotype correlations and allow comparisons. The biological processes, molecular functions with gene-interactions, and pathways that are more autism-specific may be identified through these analytical genetic studies. Currently, there are no molecular pathways known to be uniquely associated with ASD when disturbed. Some gene variants are more related to neurodevelopmental disorders and not specific for autism. Classification of gene variants that specifically cause ASD alone and not attributable to other neurodevelopmental or psychiatric disorders are under investigation as rare, large-effect mutations seen in ASD also influence cognition in a high proportion of individuals, complicating the degree of impact on the ASD phenotype vs. impact on cognitive function [42]. Certain neurodevelopmental gene variants may also impact gene function differently including neural circuits depending on an individual’s genetic background differences. 

Particular class of variants such as missense or nonsense changes may confer different effects at the protein level. Individual gene variants coding for specific amino acids may impact more important protein regions or domains with certain characteristics at specific amino acid positions, conferring mild consequences, while other amino acid positions may be more important for protein function. These areas of gene variant(s)-protein relationships will require more studies in ASD in the future using improved genetic technology, data collection, and analysis with genotype–phenotype correlations.

Brain tissue regions most often affected in ASD (e.g., hippocampus, cerebellum, etc.) may yield useful information if studied in those persons with documented autism, particularly with stored clinical and imaging data with CNVs and gene variants combined with expression patterns and methylation status in relationship to control subjects who are similarly studied. Mosaicism/tissue-specific gene expression should be considered and further studied in view of more males than females affected with ASD, particularly X-linked genes. Additionally, hormonal-mediated gender influences or differential expression in the brain should be examined for dysregulation in ASD including methylation status of brain-expressed genes on the X chromosome and interaction with autosomal genes (e.g., X-linked FMR1 gene causing fragile X syndrome [116] and CYFIP1 gene at 15q11.2 involved with coding transporter for FMR1 protein [117]). These investigations will require more specialized methods with increased sensitivity such as droplet digital PCR [118].

## 6. Summary

On behalf of individuals living with ASD and their families and for the benefit of society as a whole, increased awareness and knowledge regarding autism spectrum disorder and commonly related neurobehavioral conditions with contribution of genetic differences are imperative for healthcare professionals who provide evaluation and treatment services for ASD. Early recognition, diagnosis, and treatment should increase the likelihood that affected individuals will achieve optimal long-term outcomes and improved quality of life. Genetic and epigenetic discoveries underlying causes, as well as factors impacting treatment, such as pharmacogenetic variability, have the potential to improve the overall health of individuals with ASD. Additional clinical research to improve the evidence base for various treatment interventions for ASD with related behavioral and psychiatric challenges is desperately needed.

## Figures and Tables

**Figure 1 ijms-21-04726-f001:**
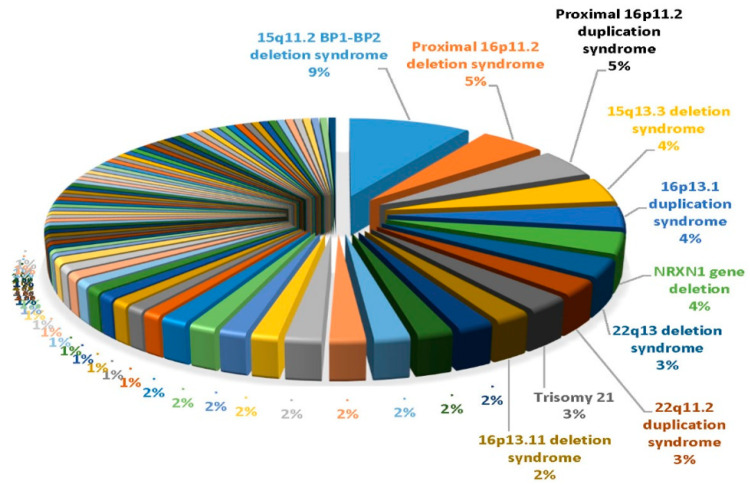
Pie chart showing the top 10 out of 85 genetic findings from data summarized by Ho et al. [47] using ultra-high-resolution chromosomal microarrays from over 10,000 consecutive patients presenting for genetic testing with neurodevelopmental disorders affecting brain function and/or structure of unknown cause with developmental/intellectual disabilities and/or ASD.

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
