# Peer review of "Clinical Assessment, Genetics, and Treatment Approaches in Autism Spectrum Disorder (ASD)"

_ijms, 2020, doi:10.3390/ijms21134726_

Round 1
Reviewer 1 Report
Thank you for asking me to review this well written manuscript that presents a summary of the status of ASD in relation to its genetic underpinning and treatment options. The text itself is well organised and easy to follow, and the references are up to date. I have a made a small number of suggestions below:
- I think you overplay the clinical genetics aspect of ASD. You mention that there is an identified (genetic) etiology in 50% which, in an unselected clinical setting, is not true. Perhaps if you included all possibly pathogenic mutations in genes that are possibly ASD-implicated you might get close to this figure, but that is stretching things. You also mention, 10% of the genetic aetiology of ASD is due to TS, and 10-20% mitochondrial. Again, when considering pathogenic variants, this is not true. There are some clinical studies that have examined diagnostic yield and the figures are substantially lower than 50%. The reality is that the vast majority of identified variants are of uncertain significance due in part to their rarity, but also the limitations of bioinformatics prediction. I would suggest a more critical and circumspect approach to the genetic literature, with perhaps some suggestions of ways forward. For example, does AI offer any solution? What about the confounding effects of clinical heterogeneity and diagnostic uncertainty? And what are the most robust findings so far?
- The section on treatment does not link well with the content up to that point; if the manuscript is principally about genetics, which it seems to be, I would suggest focusing on the implications of genetics for treatment options. You have begun to consider that in Section 4.3, but expanding this and linking back to the genetics would strengthen the manuscript.
- I also wondered: Is Figure 1 your own? If not, do you have permission to reproduce?
Author Response
Reviewer 1:
I think you overplay the clinical genetics aspect of ASD. You mention that there is an identified (genetic) etiology in 50% which, in an unselected clinical setting, is not true. Perhaps if you included all possibly pathogenic mutations in genes that are possibly ASD-implicated you might get close to this figure, but that is stretching things. You also mention, 10% of the genetic aetiology of ASD is due to TS, and 10-20% mitochondrial. Again, when considering pathogenic variants, this is not true. There are some clinical studies that have examined diagnostic yield and the figures are substantially lower than 50%. The reality is that the vast majority of identified variants are of uncertain significance due in part to their rarity, but also the limitations of bioinformatics prediction. I would suggest a more critical and circumspect approach to the genetic literature, with perhaps some suggestions of ways forward. For example, does AI offer any solution? What about the confounding effects of clinical heterogeneity and diagnostic uncertainty? And what are the most robust findings so far?
Response: We understand the comments of Reviewer 1 about the potential genetic etiology of 50% as a high number. However, heritability estimates 70-90% have been cited based on family and twin studies and several peer-reviewed published articles with 50% for potential causation. The reviewer considers this number to be an overplay in the clinical genetics aspects of ASD and we cannot dispute this statement at the current time without more advanced testing methodology, bioinformatics, and gene variant database information to produce a better understanding of genetics and ASD-specific genes as causation of ASD. Therefore, the 50% number has been changed throughout to 40%.
We also have reworded the comment regarding 10% genetic etiology for TS as TS is an example of a genetic syndrome and this statement has been altered to include other genetic syndromes such as fragile X. For mitochondrial dysfunction, supporting evidence from the literature does exist and cited but have removed the wording of causation to an association to downplay the role of genetics at this time until more research on ASD (e.g., metabolic factors and mitochondrial dysfunction) become available.
We agree that the majority of identified gene variants today are of uncertain clinical significance due to rarity or lack of information, description and understanding on involvement specifically in ASD vs neurodevelopmental disorders and intellectual disabilities requiring more studies as proposed in the Summary and Future Directions section for the readership. This included more pertinent genetic research leading to publications on these topics needed to address the genetics of ASD with impact on treatment options and plans. We also added several new references with critical circumspect approaches reported to identify more ASD-specific genes and their variants in the causation of ASD including tissue specific (e.g., brain regions) related to autism including genomic, environment, and epigenetics. We have expanded and discussed new areas of investigations as well as compounding effects of clinical heterogeneity, diagnostic uncertainty for the readership and future directions in ASD research.
The section on treatment does not link well with the content up to that point; if the manuscript is principally about genetics, which it seems to be, I would suggest focusing on the implications of genetics for treatment options. You have begun to consider that in Section 4.3, but expanding this and linking back to the genetics would strengthen the manuscript.
Response: We have added a statement and cited supporting articles in relationship to treatment and medication use including on how pharmacogenomics testing could impact on medication selection and dosage in treating features of ASD described in patients presenting with medical care in sections prior to section 4.3. We have added a linking statement in section 4.3 to the previous treatment sections to address this point. We have also added information in the new Summary and Future Directions section about patient care and treatment with interplay of pharmacogenetics as a representative of implications of genetics to treatment. It is also understood that genetic syndromes such as fragile X, Prader-Willi, Down and Rett syndromes can present with ASD but have different genetic causes (i.e., triplet repeat mutations, imprinting and methylation disturbances at single gene to multiple gene level, chromosomal numerical or structural defects) possess a variable list of psychiatric/ behavioral features which will require different treatment approaches- all based on their clinical genetics status and diagnosis.
I also wondered: Is Figure 1 your own? If not, do you have permission to reproduce?
Response: Regarding figure 1, we used data from the cited report in which I am the senior author of the ultrahigh microarray analysis performed by a commercial CLIA approved laboratory in over 10,000 consecutive patients presenting for genetic services with features of autism. This pie chart which has not been published Illustrates the more common microarray findings in this patient population with 15q11.2 BP1-BP2 deletion (Burnside-Butler) syndrome as the most common microarray result identified. Hence, I have cited this report in the literature and the figure legend giving full credit to the initial source and have not impinged on copy right issues as the original article was published in an open access source.
Reviewer 2 Report
Thank you for the opportunity to review the manuscript Clinical Assessment, Genetics, and Treatment Approaches in Autism Spectrum Disorder. This is a huge topic and the authors strive to summarize a very large body of literature into a readable format. There is lots of important information throughout the manuscript however overall, the paper feels already dated. Most of the references, in a quickly updating field, are from before 2015 or so. In addition, some sections seem very briefly glossed over (understandably) but how they read are more definitive. For example, stating that the ADOS and the ADI are the gold standards is true from a research perspective but in clinical operations the ADI for example is rarely done. This leads more to the question on who this manuscript is intended for. it would be a shame if clinical geneticists or others who are reading a clinical diagnostic evaluation dismiss it if not perceived as gold standard without the use of the ADI. This is particularly highlighted as the section reads as what needs to be done (3 generation family history, etc.) but then also includes the ADOS and ADI-R. In addition, minor feedback is to list the ADOS as the ADOS-2.
The early introduction paragraphs also read very piecemeal which thus make it difficult to follow. Recommend reorganizing this section. In addition, I recommend checking the AAP recommendations as it is my understanding that it is broadband screening for any developmental concerns at 9, 12 and 15 months and autism specific screenings in addition and 18 and 24 months. In this case, the 12 month screening for ASD would be misleading as it is extremely rare for a clinician to diagnose autism at this age, with the earlier stability studies leaning more towards 15-18 months at the younger ranges.
It is unclear why the first row in table 1 is bolded. Is something supposed to be standing out about Fragile X and Apert syndrome?
The treatment section break down the initial into children and adolescents then adults and cites a paper from 2014 for the child/adolescent. The field has also come along way since then for behavioral treatments so I would recommend either updating or stating as in particular the whole early intervention is missing here.
The summary could also use some increased depth.
Author Response
Thank you for the opportunity to review the manuscript Clinical Assessment, Genetics, and Treatment Approaches in Autism Spectrum Disorder. This is a huge topic and the authors strive to summarize a very large body of literature into a readable format. There is lots of important information throughout the manuscript however overall, the paper feels already dated. Most of the references, in a quickly updating field, are from before 2015 or so. In addition, some sections seem very briefly glossed over (understandably) but how they read are more definitive. For example, stating that the ADOS and the ADI are the gold standards is true from a research perspective but in clinical operations the ADI for example is rarely done. This leads more to the question on who this manuscript is intended for. it would be a shame if clinical geneticists or others who are reading a clinical diagnostic evaluation dismiss it if not perceived as gold standard without the use of the ADI. This is particularly highlighted as the section reads as what needs to be done (3 generation family history, etc.) but then also includes the ADOS and ADI-R. In addition, minor feedback is to list the ADOS as the ADOS-2. \
Response: Additional searches have been completed and more recent references have been added to the manuscript. The term “gold standard” in terms of the ADOS and ADI has been removed as the reviewer is correct that these instruments are not required nor routinely used in clinical practice. ADOS was updated to the current ADOS-2, with updated reference noted for this instrument.
The early introduction paragraphs also read very piecemeal which thus make it difficult to follow. Recommend reorganizing this section. In addition, I recommend checking the AAP recommendations as it is my understanding that it is broadband screening for any developmental concerns at 9, 12 and 15 months and autism specific screenings in addition and 18 and 24 months. In this case, the 12 month screening for ASD would be misleading as it is extremely rare for a clinician to diagnose autism at this age, with the earlier stability studies leaning more towards 15-18 months at the younger ranges.
Response: The introduction paragraph was revised, updated and reorganized. The reviewer is correct about AAP recommendation for ASD screening at 18 months so this correction was made. We have expanded and further clarified with an effort to improve the manuscript on the topic of genetics and treatment approaches in ASD including pharmacogenetics.
It is unclear why the first row in table 1 is bolded. Is something supposed to be standing out about Fragile X and Apert syndrome?
Response: This has been corrected. Thank you for identifying this error.
The treatment section break down the initial into children and adolescents then adults and cites a paper from 2014 for the child/adolescent. The field has also come a long way since then for behavioral treatments so I would recommend either updating or stating as in particular the whole early intervention is missing here.
Response: Updated information and references were added to the section on Behavioral Interventions in Children and Adolescents for more completeness.
The summary could also use some increased depth.
Response: The summary has been expanded and further directions added to the title on the topic of genetics, autism and treatment approaches for this readership.
Thank you for allowing us to revise the manuscript as described above and required to strengthen the manuscript prior to publication.
Round 2
Reviewer 1 Report
Thank you for asking me to take another look at this manuscript.
1. Regarding my first point, I don't dispute that heritability indicates a large genetic component for ASD, but the reality is that we have only identified a very small proportion of this risk, and "potential causation" is a far cry from actually identifying the causal genes. the reality of translational genetics in ASD is far less optimistic than is alluded to in this manuscript. This is important to consider critically, i.e. why is that? I am not suggesting this point is discussed at length in the paper, but it is important to be balanced about the status of clinical genetics in ASD.
2. Regarding heritability, why have you not also discussed the papers that find a much lower heritability. These (e.g Hallmeyer et al) are also relevant.
3. On p. 5, "over 800 genes associated with" should read "implicated in". Association carries a very specific meaning.
4. On p. 5, it might be clearer to write "neurodevelopmental disorders can cumulatively affect ...". Can you expand on this figure: ASD, say, 1.5%, ID 1%, what others were you referring to?
5. You also mention "20-25%" with macrocephaly. If you were to take a random clinic samle of children with ASD, you would not find 20% with macrocephaly. Again, you are quoting inflated figures.
Author Response
Regarding my first point, I don't dispute that heritability indicates a large genetic component for ASD, but the reality is that we have only identified a very small proportion of this risk, and "potential causation" is a far cry from actually identifying the causal genes. the reality of translational genetics in ASD is far less optimistic than is alluded to in this manuscript. This is important to consider critically, i.e. why is that? I am not suggesting this point is discussed at length in the paper, but it is important to be balanced about the status of clinical genetics in ASD.
Response: We appreciate the reviewer's dedication to research in identifying causative factors contributing to autism and the complexity of the genetics of this disorder(s). We have added new references about the genetics of autism and that risk affects are highly variable and relate to other clinical conditions besides autism making it difficult to find ASD-specific gene variants as gene variants may impact in common biological pathways or interactions. To address the current challenges translating autism genetics into clinical practice more research is needed to identify genetic etiology and pathogenesis of ASD which remain largely unclear.
Regarding heritability, why have you not also discussed the papers that find a much lower heritability. These (e.g Hallmeyer et al) are also relevant.
Response: We have noted that autism is considered the most heritable neurodevelopmental disorder based on a large difference in concordance rates or heritability estimates between monozygotic and dizygotic twins with monozygotic twins having rates that are nearly 3 times higher than rates found in dizygotic twins (Hallmayer et al. 2011). They also concluded that susceptibility to ASD shows moderate genetic heritability and substantially shared environmental components based on studies indicating a challenge to find genetic causation for autism. Also, a comment has been made in the first paragraph of 2. Diagnosis and Genetics of ASD section that the anticipated number of pathogenic variants have not been identified to date. This information has been added to the text.
On p. 5, "over 800 genes associated with" should read "implicated in". Association carries a very specific meaning.
Response: We have changed 'associated' to implicated' in relationship to the 800 genes identified in the literature regarding ASD. We agree that this change was needed in the text and used accordingly throughout.
On p. 5, it might be clearer to write "neurodevelopmental disorders can cumulatively affect ...". Can you expand on this figure: ASD, say, 1.5%, ID 1%, what others were you referring to?
Response: We have modified the statement on page 5 regarding neurodevelopmental disorders and have expanded the legend of the figure to further clarify data reported by Ho et al. used microarray analysis in over 10,000 individuals presenting for genetic services and testing. In the study, several individuals with microdeletions were found indicating the presence of classical clinical genetic disorders (e.g., DiGeorge syndrome) with neurodevelopmental problems including intellectual disabilities and/or autism in the study participants.
You also mention "20-25%" with macrocephaly. If you were to take a random clinic sample of children with ASD, you would not find 20% with macrocephaly. Again, you are quoting inflated figures.
Response: We have removed the comment regarding 20-25% with macrocephaly. We have added that large appearing head size is common in children with autism. We also note that mutations of the PTEN tumor suppressor gene are seen in children with autism and extreme macrocephaly to avoid confusion.
The changes made in the manuscript can be seen on lines 58-60, 62, 98-101, 112-127, 131, 175, 230-232, 267-268, 301-302, 305, and 479-483.

Reviewer 2 Report
The authors have revised the manuscript according to the reviewer recommendations and I do not have any additional comments at this time.
Author Response
No changes have been made from the first round of reviews.